# Improved Color Purity of Monolithic Full Color Micro-LEDs Using Distributed Bragg Reflector and Blue Light Absorption Material

**Shao-Yu Chu [1], Hung-Yu Wang [1], Ching-Ting Lee [1,2], Hsin-Ying Lee [1,\*] , Kai-Ling Laing [3], Wei-Hung Kuo [3], Yen-Hsiang Fang [3] and Chien-Chung Lin [3,4]**

1   Department of Photonics, National Cheng Kung University, Tainan 701, Taiwan; kevinvicky168@gmail.com (S.-Y.C.); aodhan20015@gmail.com (H.-Y.W.); ctlee@ee.ncku.edu.tw (C.-T.L.)
2   Department of Electrical Engineering, Yuan Ze University, Taoyuan 320, Taiwan
3   Electronic and Optoelectronic System Research Laboratories, Industrial Technology Research Institute, Hsinchu 310, Taiwan; KL@itri.org.tw (K.-L.L.); GuoWeiHong@itri.org.tw (W.-H.K.); YHFang@itri.org.tw (Y.-H.F.); chienchunglin@faculty.nctu.edu.tw (C.-C.L.)
4   Institute of Photonic System, National Chiao Tung University, Tainan 711, Taiwan
\*   Correspondence: hylee@ee.ncku.edu.tw; Tel.: +886-6-2082368

**Abstract:** In this study, CdSe/ZnS core-shell quantum dots (QDs) with various dimensions were used as the color conversion materials. QDs with dimensions of 3 nm and 5 nm were excited by gallium nitride (GaN)-based blue micro-light-emitting diodes (micro-LEDs) with a size of 30 μm × 30 μm to respectively form the green and red lights. The hybrid Bragg reflector (HBR) with high reflectivity at the regions of the blue, green, and red lights was fabricated on the bottom side of the micro-LEDs to reflect the downward light. This could enhance the intensity of the green and red lights for the green and red QDs/micro-LEDs to 11% and 10%. The distributed Bragg reflector (DBR) was fabricated on the QDs color conversion layers to reflect the non-absorbed blue light that was not absorbed by the QDs, which could increase the probability of the QDs excited by the reflected blue light. The blue light absorption material was deposited on the DBR to absorb the blue light that escaped from the DBR, which could enhance the color purity of the resulting green and red QDs/micro-LEDs to 90.9% and 90.3%, respectively.

**Keywords:** blue light absorption material; color conversion layer; distributed Bragg reflector; hybrid Bragg reflector; micro-light-emitting diodes; quantum dots

## 1. Introduction

In recent years, display screens have been widely used in daily life, and liquid crystal display (LCD) and organic light-emitting diode (OLED) displays are the mainstream of display technology [1]. However, breakthrough progress has been made in semiconductor manufacturing technology, leading to the abrupt rise of micro-light-emitting diodes (micro-LEDs). In particular, the micro-LEDs have advantages of long lifetime, high luminous efficiency, and smaller volume [2–4]. On the other hand, micro-LEDs can effectively reduce energy consumption and improve pixel characteristics, which can be expected to become the mainstream of next-generation display technology. In general, each pixel of the full color micro-LED display is constructed by red, green, and blue light sources; then different light source colors can be obtained by controlling the ratio of the three primary colors. However, this method includes disadvantages such as different lifetime and complex driver circuits.

To overcome the above-mentioned problem, the blue light emitted from the gallium nitride (GaN)-based micro-LEDs was used to excite the quantum dots (QDs) with various dimensions to form

the monolithic red and green QDs/micro-LEDs [5–7]. In this work, the QDs were filled in the regions around the black photoresist with very low transmittance in the visible light region. The function of the black photoresist is to prevent the light emitted from the sidewall of the micro-LEDs to reduce the crosstalk between the micro-LEDs. However, since the excitation efficiency of the red and green QDs was still low, a large amount of non-absorbed blue light would respectively blend with the green light and red light and be simultaneously emitted from the green and red QDs/micro-LEDs. Consequently, the color purity of red and green QDs/micro-LEDs would be not good. The distributed Bragg reflector (DBR) with high reflectivity in the blue light region was deposited on the QD color conversion layer to solve the problem of the non-absorbed blue light [8,9]. However, since the DBR reflectivity depends on the incident angle of the light, the DBR reflectivity will be reduced if the light is not incident perpendicularly into the DBR structure [10,11]. Consequently, in this study, the blue light absorption layer was utilized and deposited on the DBR to absorb the escaped blue light from the DBR, which could improve the color purity of the monolithic full color micro-LEDs. To further enhance the output light intensity, a hybrid Bragg reflector (HBR) was deposited on the bottom side of the micro-LEDs to reflect the downward propagation of red, green, and blue lights.

## 2. Materials and Methods

### 2.1. Materials

In this work, the epitaxial wafers of the GaN-based blue micro-LEDs were supported by Epistar Co., Hsinchu, Taiwan. CdSe/ZnS core-shell quantum dots with an average dimension of 5 nm and 3 nm were purchased from Taiwan Nanocrystal Inc., Tainan, Taiwan. The granules of titanium dioxide ($TiO_2$) (99.9%) and silicon dioxide ($SiO_2$) (99.999%) were purchased from Admat Inc., Pennsylvania, PA, USA. The black photoresist (model: ABK408X) was supported by Daxin Materials Co., Taichung, Taiwan. The blue light absorption material (model: Eusorb UV-1995) was supported by Eutec Chemical Co., Taipei, Taiwan.

### 2.2. Experimental Procedure

Figure 1a illustrates the schematic configuration of the monolithic full color micro-LEDs. A metal organic chemical vapor deposition system was used to epitaxy the GaN-based blue micro-LEDs structure on c-plane sapphire substrates. The structure of the micro-LEDs was constructed by a 2.8 μm-thick GaN buffer layer, a 4 μm-thick *n*-GaN layer, an undoped InGaN/GaN (3/7 nm, 10 pairs) multiple quantum wells (MQW) active layer, a 33 nm-thick *p*-AlGaN layer, and a 150 nm-thick *p*-GaN layer. A 500 nm-thick Ni metal was deposited on the *p*-GaN layer as the metal mask to protect the mesa (30 μm × 30 μm) of the micro-LEDs through an electron-beam evaporator. After the mesa dry etching, the remaining Ni metal was removed by using aqua regia. The Ti/Al/Pt/Au (25/100/50/150 nm) metals were deposited on the patterned *n*-GaN layer by an electron-beam evaporator and then annealed in a pure $N_2$ environment at 850 °C for 2 min using a rapid temperature annealing (RTA) system to obtain *n*-electrode ohmic contact [12]. Afterward, the thin Ni/Au (3/3 nm) metals were deposited on the *p*-GaN surface treated by $(NH_4)_2S_x$ solution (S = 6%) for 30 min [13] as the current spreading layer, and the Ni/Au (20/100 nm) metals were deposited as the *p*-electrode. To obtain the *p*-electrode ohmic contact, the samples were annealed in an air environment at 500 °C for 10 min using the RTA system. In this work, an extended *p*-electrode pattern was designed to facilitate electrical measurement of the micro-LEDs (as shown in Figure 1b). The 2.5 μm-thick black photoresist was spun on the samples and patterned between the monolithic full color micro-LEDs through a photolithography method. The black photoresist was solidified through the RTA system in a $N_2$ environment at 150 °C for 10 min. Since black photoresist had a low average transmittance of 0.56% in the visible light region, it could prevent crosstalk among the full color micro-LEDs. Subsequently, the HBR composed of $(TiO_2/SiO_2)^{m\text{-}pairs}$ (48/77 nm) and a Ag (200 nm) metal reflector was deposited on the bottom side of the samples through the electron beam evaporator. The red (green) CdSe/ZnS core-shell QDs with

an average dimension of 5 nm (3 nm) and the poly(methyl methacrylate) (PMMA) were added into toluene, and then the red (green) QD slurry was filled into the openings in the black photoresist layer to form the red (green) QD color conversion layer. Subsequently, the electron beam evaporator was used to alternately deposit $(TiO_2/SiO_2)^{n-pairs}$ (48/77 nm)/$TiO_2$ (48 nm) on the red (green) QDs at 100 °C as the DBR structure. Finally, a 100 nm-thick blue light absorption layer was formed on the DBR by spin-coating.

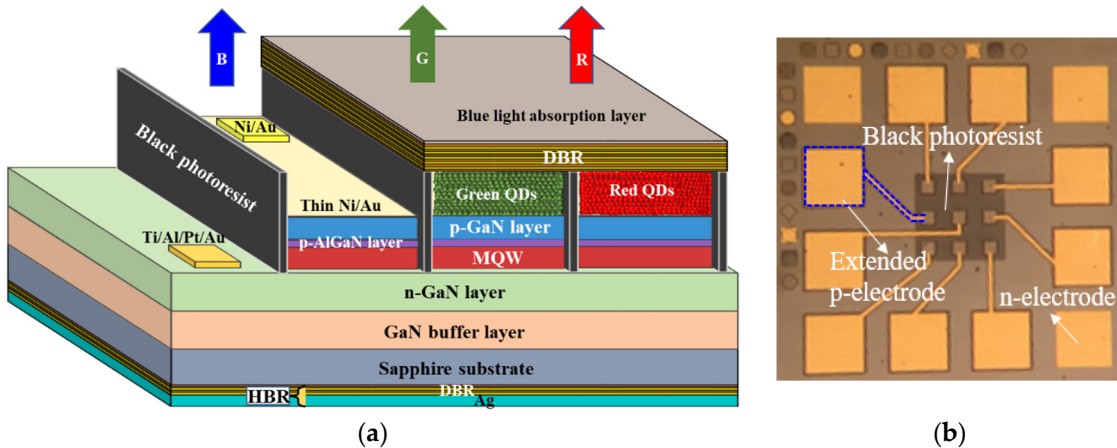

**Figure 1.** (**a**) The schematic configuration of the monolithic full color micro-light-emitting diodes (micro-LEDs). (**b**) Top view photograph of the blue micro-LEDs.

## 3. Results and Discussion

Figure 2a and the inset figure show the electroluminescence (EL) spectrum and the current–voltage (*I–V*) characteristics of the blue micro-LEDs, respectively. In this work, an EL measurement system was installed in a black box and equipped with an Agilent 4156C semiconductor parameter analyzer, an optical fiber, and a spectrometer (Ocean Optics USB2000, Florida, FL, USA) system. In the EL measurement process, the resulting samples were placed on the stage in the black box and the optical fiber was fixed above the micro-LEDs with a distance of 1 mm to catch the light emitted from the micro-LEDs. The maximum EL intensity of all the micro-LEDs at the applied current of 1.2 mA was measured by adjusting the sample stage. The emission wavelength and the threshold voltage of the blue micro-LEDs was 451 nm and 2.5 V, respectively. The emission wavelengths of 625 and 550 nm for the red and green QDs excited by a tunable laser with a wavelength of 451 nm were obtained, respectively (Figure 2b). The full width at half maximum (FWHM) of the blue, green, and red lights emitted from the blue micro-LEDs, red and green QD micro-LEDs was 27.0, 32.2, and 34.5 nm, respectively.

Generally, the QD density in the color conversion layer would affect the color conversion efficiency of the QDs. Consequently, the various weight ratios (10:20, 10:10, 20:10, and 40:10 mg, hereafter referred to as 1:2, 1:1, 2:1, and 4:1) of the green and red QDs:PMMA were added into toluene (1 mL) to form the QD slurry with various QD densities. Figure 3a,b present the emission spectra of various green QDs:PMMA ratios and red QDs:PMMA ratios excited at the tunable laser with a wavelength of 451 nm. As shown in Figure 3, the emission intensity of the green QDs:PMMA = 1:1 and red QDs:PMMA = 1:1 was larger than that of the other green and red QDs:PMMA ratios. This phenomenon can be attributed to the excessive QDs in the QDs slurry with QDs:PMMA ratios of 2:1 and 4:1, which would reabsorb the converted green and red lights, reducing the output intensity of the converted green and red lights [14]. In contrast, the fewer QDs in the QD slurry with the QDs:PMMA ratio of 1:2 would lead to the absorbed amount of the blue light being too low. Consequently, the emission intensity of the QD slurry with QDs:PMMA ratio of 1:2 was smaller in comparison with the QD slurry with the QDs:PMMA ratio of 1:1.

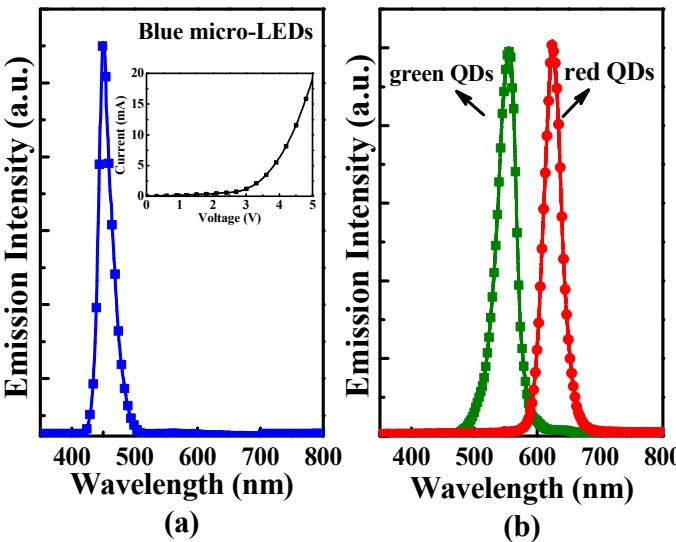

**Figure 2.** (**a**) The electroluminescence (EL) spectra of the GaN-based blue micro-LEDs operated at a bias of 3 V. The inset figure shows the current–voltage characteristics. (**b**) The emission spectrum of the green and red QDs.

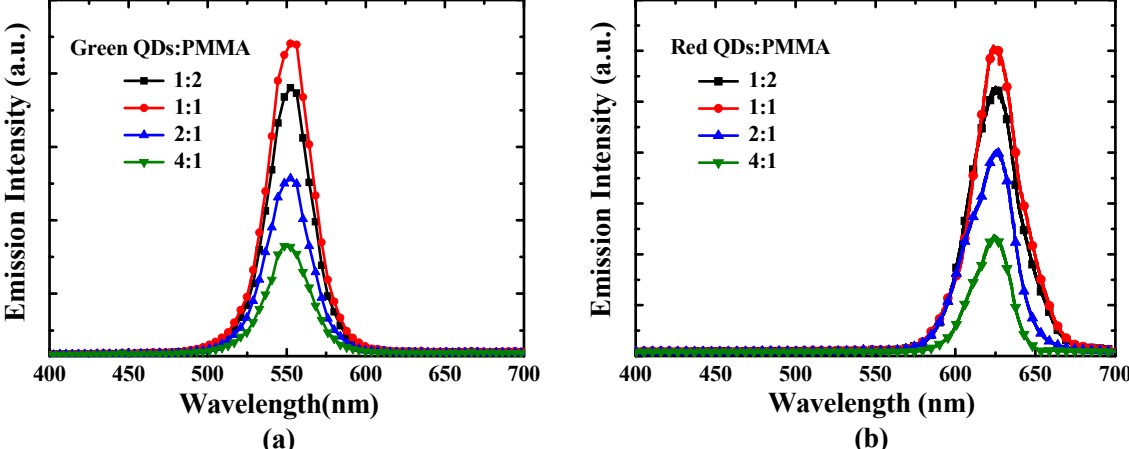

**Figure 3.** The emission spectra of various (**a**) green core-shell quantum dots: poly (methyl methacrylate) (QDs:PMMA) ratios and (**b**) red QDs:PMMA ratios excited by a tunable laser with a wavelength of 451 nm.

To improve the output light intensity of the monolithic full color micro-LEDs, the HBR with high reflectivity for the red, green, and blue lights was deposited on the bottom side of the samples. The HBR was constructed by $(TiO_2/SiO_2)^{m\text{-}pairs}$ (48/77 nm) and Ag (200 nm) metal. Figure 4 shows the reflectivity spectra of the Ag metal and $(TiO_2/SiO_2)^{m\text{-}pairs}$ ($m$ = 1, 2, and 3)/Ag measured using an UV–Visible spectrophotometer. The reflectivity of all of the $(TiO_2/SiO_2)^{m\text{-}pairs}$/Ag HBR structures was higher than that of the single Ag metal. Furthermore, the highest reflectivity of $(TiO_2/SiO_2)^{2\text{-}pairs}$/Ag was 97.5%, 98.9%, and 98.8% at the wavelengths of 451, 550, and 625 nm, respectively. Consequently, the downward propagation of the red, green, and blue lights could be effectively reflected by the HBR, which could improve the output light intensity.

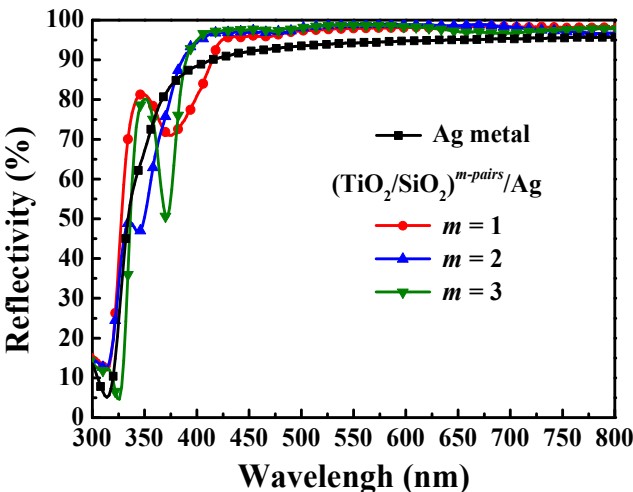

**Figure 4.** The reflectivity spectra of the Ag metal and $(TiO_2/SiO_2)^{m-pairs}$ ($m$ = 1, 2, and 3)/Ag.

In this work, the green and red QD color conversion layers were used to convert the blue light into the green and red lights, respectively. To avoid the red and green lights that emitted from the QDs/micro-LEDs blended with the non-absorbed blue light, the DBR with high reflectivity at the wavelength of 451 nm should be deposited on the red and green QDs color conversion layers. Simultaneously, the DBR with high transmittance at wavelengths of 550 nm and 625 nm was requested. The DBR was composed of $(TiO_2/SiO_2)^{n-pairs}$ (48/77 nm)/$TiO_2$ (48 nm). Figure 5 presents the reflectivity spectra of the $(TiO_2/SiO_2)^{n-pairs}$ ($n$ = 2, 3, 4, and 5)/$TiO_2$ DBR structure measured using an UV–Visible spectrophotometer. The number of $n$-pairs would obviously affect the reflectivity of the DBR at the blue light range. The DBR reflectivity at the wavelength of 451 nm increased with an increase in the number of $n$-pairs. Although the highest DBR reflectivity at the wavelength of 451 nm was 98.6% as $n$ = 6, the DBR reflectivity at wavelengths of 550 and 625 nm was 31.9% and 15.8%, respectively, which adversely affected the emitting of the green light. The reflectivity of the $(TiO_2/SiO_2)^{5-pairs}$/$TiO_2$ DBR structure at the wavelengths of 451, 550, and 625 nm was 96.4%, 13.7%, and 7.6%, respectively. Consequently, the 5-pairs $(TiO_2/SiO_2)$ were more suitable for the DBR structure than the 6-pairs $(TiO_2/SiO_2)$.

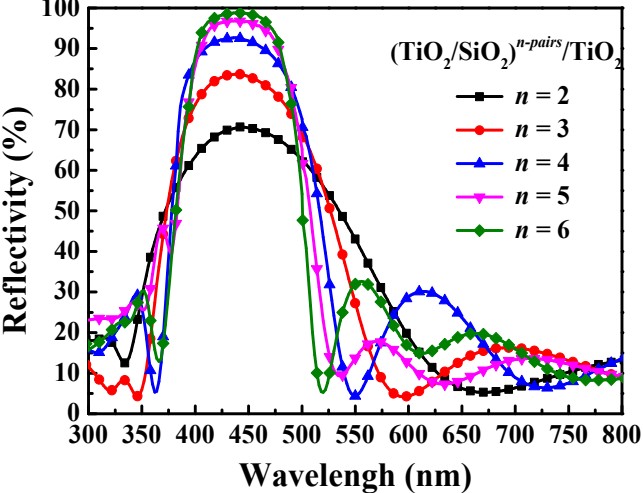

**Figure 5.** The reflectivity spectra of the $(TiO_2/SiO_2)^{n-pairs}$ ($n$ = 2, 3, 4, and 5)/$TiO_2$ distributed Bragg reflector (DBR) structure.

The EL spectra of the resulting blue micro-LEDs and the resulting QDs/micro-LEDs were measured to verify the function of the HBR, DBR and blue light absorption layer. Figure 6 shows the EL spectra of

the resulting green and red QDs/micro-LEDs and the blue micro-LEDs with HBR. As shown in Figure 6, the output intensity of the green and red QDs/micro-LEDs with HBR was respectively increased 11% and 10% in comparison with the QDs/micro-LEDs without HBR. This proved that the HBR could reflect the downward emitted red, green, and blue lights. Furthermore, it was worth noting that only part of the blue light was absorbed by the green and red QDs, which leads to non-absorbed blue light blending with the green light and red light simultaneously being emitted from the green and red QDs/micro-LEDs with HBR. Therefore, the non-absorbed blue light emitted from the QDs/micro-LEDs with HBR could be effectively reflected back into the QD color conversion layer by using the DBR with high reflectivity at the wavelength of 451 nm and excite more red and green lights. Compared with the QDs/micro-LEDs with HBR and without DBR, the output intensity of the green and red QDs/micro-LEDs with HBR and DBR was increased 20% and 23%, respectively. However, since the reflectivity of the DBR was not 100% and the reflectivity of the DBR was decreased as the light was not perpendicularly incident into the DBR structure, a small portion of blue light still escaped, which would affect the color purity of the QDs/micro-LEDs. In this work, the blue light absorption layer was deposited on the DBR to reduce the escaped blue light. The absorptivity spectrum of the blue light absorption layer is shown in Figure 7. The absorptivity of the blue light absorption layer at wavelengths of 451, 550, and 625 nm was 77.2%, 1.3%, and 1.8%, respectively. Since the blue light escaped from the DBR could be absorbed by the blue light absorption layer, the intensity of the blue light for the QDs/micro-LEDs was reduced, as shown in Figure 6, which could improve the red and green color purity.

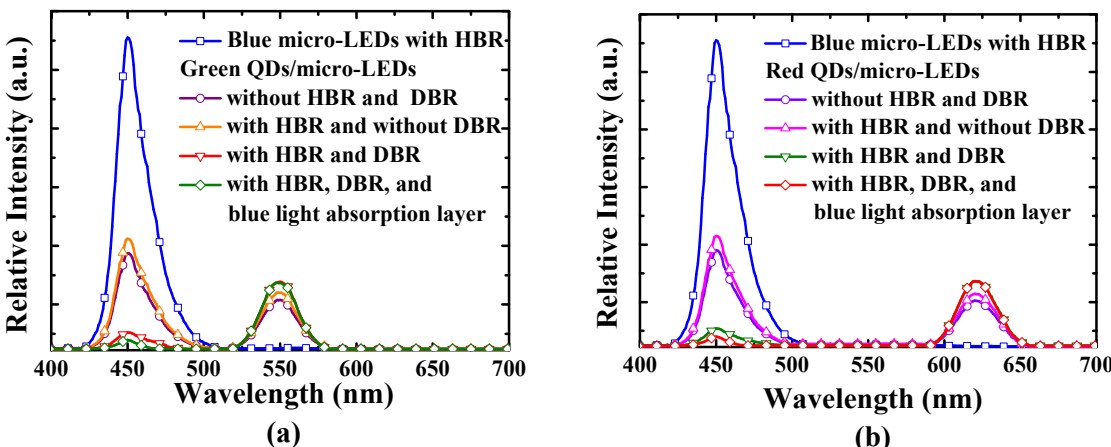

**Figure 6.** The EL spectra of the resulting (**a**) green QDs/micro-LEDs and (**b**) red QDs/micro-LEDs.

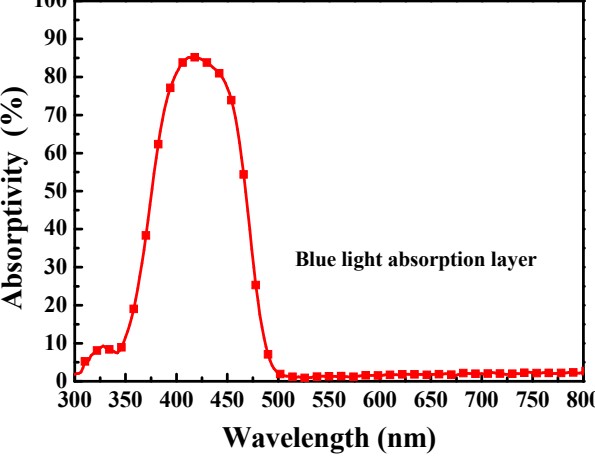

**Figure 7.** The absorptivity spectrum of the blue light absorption layer.

Figure 8 shows the international commission on illumination (CIE) chromaticity coordinates of the resulting micro-LEDs. Compared with the green and red QDs/micro-LEDs with HBR and without DBR, the CIE chromaticity coordinate of the green and red QDs/micro-LEDs with HBR and DBR shifted from (0.211, 0.221) to (0.271, 0.531) and from (0.311, 0.109) to (0.564, 0.226), respectively. The CIE chromaticity coordinate of the green and red QDs/micro-LEDs with HBR, DBR, and blue light absorption layer was (0.292, 0.632) and (0.646, 0.279), respectively. The color purity of the LEDs could be estimated by Equation (1) as follows [15–17]:

$$\text{color purity} = \frac{\sqrt{(X - X_i)^2 - (Y - Y_i)^2}}{\sqrt{(X_d - X_i)^2 - (Y_d - Y_i)^2}} \times 100\% \tag{1}$$

where ($X$, $Y$) is the CIE chromaticity coordinate of the resulting QDs/micro-LEDs, ($X_i$, $Y_i$) is the CIE chromaticity coordinate of the blue micro-LEDs, ($X_d$, $Y_d$) is the CIE chromaticity coordinate of green light and red light at color purity of 100%. The CIE chromaticity coordinate of the green light with wavelength of 550 nm and the red light with wavelength of 625 nm was (0.301, 0.692) and (0.701, 0.299), respectively. Compared with the green and red QDs/micro-LEDs with HBR and without DBR, the color purity of the green and red QDs/micro-LEDs with HBR and DBR was improved from 28.2% to 75.4% and from 27.7% to 74.2%, respectively. Consequently, the DBR could effectively reflect the non-absorbed blue light to improve the color purity. The color purity of the green and red QDs/micro-LEDs with HBR, DBR, and blue light absorption layer was further improved to 90.9% and 90.3%, respectively.

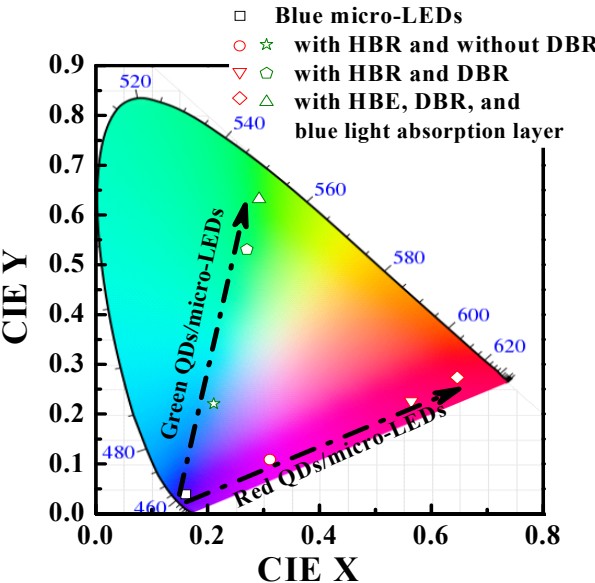

**Figure 8.** The international commission on illumination (CIE) chromaticity coordinates of the resulting micro-LEDs.

## 4. Conclusions

In this study, monolithic full color micro-LEDs were successfully fabricated by using blue micro-LEDs to excite various dimensions of CdSe/ZnS core-shell QDs. To increase the output intensity and improve the color purity, HBR and DBR were used as the bottom reflector for the full colors and the top reflector for the blue light, respectively. The downward emitted full colors could be reflected by the bottom reflector HBR, which could increase the probability of blue light exciting the red and green QDs. Consequently, the output intensity of the green and red QDs/micro-LEDs with HBR was respectively increased 11% and 10% in comparison with the QDs/micro-LEDs without HBR. Furthermore, the output intensity of the green and red QDs/micro-LEDs with HBR and DBR was

further increased 20% and 23%, respectively, as the blue light reflected by the DBR could excite QDs to form more red and green lights. Finally, the blue light absorption layer with absorptivity of 77.2% at the wavelength of 451 nm was developed to absorb the blue light that escaped from the DBR. The color purity of the green and red QDs/micro-LEDs with HBR, DBR, and blue light absorption layer was further improved to 90.9% and 90.3%, respectively.

**Author Contributions:** Conceptualization, H.-Y.L., C.-T.L., K.-L.L., W.-H.K., Y.-H.F., and C.-C.L.; Data curation, S.-Y.C. and H.-Y.W.; Funding acquisition, H.-Y.L., C.-T.L., K.-L.L., W.-H.K., Y.-H.F., and C.-C.L.; Investigation, S.-Y.C., H.-Y.W., H.-Y.L., and C.-T.L.; Writing—original draft, S.-Y.C. and H.-Y.W.; Writing—review and editing, H.Y.L. All authors have read and agreed to the published version of the manuscript.

**Funding:** This work was supported by the Ministry of Science and Technology (MOST), Taiwan under Nos. MOST-108-2221-E-006-196-MY3 and MOST 108-2221-E-006-215-MY3, and the Industrial Technology Research Institute, Taiwan.

**Conflicts of Interest:** The authors declare no conflict of interest.

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
