# Peer review of "Improved Color Purity of Monolithic Full Color Micro-LEDs Using Distributed Bragg Reflector and Blue Light Absorption Material"

_coatings, doi:10.3390/coatings10050436_

Round 1
Reviewer 1 Report
The work is well written and suitable for publication in MDPI.
For a more comprehensive understanding I would suggest to detail more the introduction (longer one) as well as adding more reference.
Thanks reviewer’s commons. We added the sentences “In this work, the QDs were filled in the regions around the black photoresist with low transmittance in the visible light region. The function of the black photoresist could prevent the light emitting from the sidewall of the micro-LEDs to reduce the crosstalk between the micro-LEDs.” in p. 2, line 44-47 in the revised manuscript. The two references [10,11] were also added in p. 2, line 54 and p. 8, line 235-240 in the revised manuscript.
Reviewer 2 Report
The authors in this current work investigated multiple methods to improve the color purity of micro-LEDs. Among those methods, hybrid bragg reflector (HBR) and distributed bragg reflector (DBR) have both been reported in an analogous work made by the same authors. (Chen, Guan-Syun, et al. IEEE Photonics Technology Letters 30.3 (2017): 262-265.) Using Eusorb UV-1995 as a blue light absorbing layer indeed is a new method, but doesn’t present enough originality. Overall, major revisions need to be done before being considered for publishing.
1. The main originality of the present work is to improve color purity by using a blue light absorbing material. However, the authors haven’t given enough information about this material. What is the chemical formula of this material? What applications has it been used for?
The blue light absorbing material (Model: Eusorb UV-1995) was supported by Eutec Chemical Co., Taiwan. We contacted with company to check the chemical formula of the blue light absorbing material. Since the applied patent of the blue light absorbing material is being processed, please allow us do not show the chemical formula of the blue light absorption material. Currently, the blue light absorption material is coated on the glass and the plates as the phone protection, which can protect the eyes.
2. Since both HBR and DBR structures have already been used in previous work, the authors may consider an optimization of those structures. For example, how is the thickness of TiO2/SiO2 pair determined? Can the thickness be further optimized?
(a)The thickness of the DBR mirror could be initially calculated by the formula as follows: ?=?/4? (1)where d and n are the thickness and refractive index of TiO2 and SiO2 layers, respectively, λ is the wavelength of the reflected wave. We used the calculated thickness of TiO2 and SiO2 layers as the reference to prepared the DBR structure with high reflectivity at blue light. By slightly adjusting the thickness of TiO2 and SiO2 layers, the (TiO2/SiO2)5-pairs/TiO2 DBR structure at the wavelength of 451 nm, 550 nm, and 625 nm was 96.4%, 13.7%, and 7.6%, respectively. The thickness of TiO2 and SiO2 measured by alpha-step surface profier was 48 nm and 77 nm, respectively.
(b) In this work, the thickness of TiO2 and SiO2 layers was further optimized to find the DBR structure with high reflectivity at blue light.
3. The authors also need to give more details about the experimental setup. For example, what instrument did you use to measure the electroluminescence/emission spectra? What is the measurement condition? Were they all measured at the same current density?
In this work, an electroluminescence (EL) system was installed in a black box and was equipped with an Agilent 4156C semiconductor parameter analyser, an optical fiber and spectrometer (Ocean Optics USB2000) system. In the EL measurement process, the resulting samples were placed on the stage in the black box and the optical fiber was fixed above the
micro-LEDs with a distance of 1 mm to catch the emitting light from the micro-LEDs. The maximum EL intensity of all the micro-LEDs applied current of 1.2 mA was measured by adjusted the sample stage.
4. Are those reflectivity spectra in Figure 4 and 5 measured results or simulated results? If measured results, how are they measured? Do the results match theoretical values?
The reflectivity spectra of DBR and HBR were measured using an UV-Visible spectrophotometer and shown in Figure 4 and Figure 5, respectively. The measurement results were matching with the theoretical values. The measurement equipment was added in p. 4, line 126-128 and p. 5, line 139-141 in the revised manuscript.
5. The language in the present manuscript requires careful proofreading. For example, the section titles should be Experimental instead of Manufacturing instead of Results and discussions.
Thanks reviewer’s comments. The section titles were revised in p. 2, line 67 and p. 3, line 96 in the revised manuscript.
Reviewer 3
The manuscript presents some interesting results about a device configuration to improve quantum dot emission efficiency using a Bragg reflector.
Nevertheless, it has severe flaws in presentation, either as concerning language, or from the point of view of scientific evidence.
As an example: results presented in Figure 6 are called to support a claim in the text, but there is no relationship between Figure 6 and the claimed result in its comment.
There is a lack of information and comment about the role played by the structure of HBR and the choice of the layer number of DBR is not supported by any physical consideration.
The hybrid Bragg reflector (HBR) was composed with (TiO2/SiO2)2-pairs (48/77 nm) and a 200-nm-thick Ag metal. The functions of the HBR included high reflectivity at the regions of the blue, green, and red lights, and high reflectivity at omnidirectional. Consequently, the HBR deposited on the bottom side of the micro-LEDs to reflect the downward blue, green, and red lights, which could improve the output light intensity.
A distributed Bragg reflector (DBR) is a structure formed from multiple layers of alternating materials with varying refractive index. According to the reflectivity of the formula for DBR, the reflectivity of the DBR was increase with an increase of the number of repeated pairs of low/high refractive index material. In this work, the DBR was fabricated by alternately depositing (TiO2/SiO2)n-pairs (48/77 nm)/TiO2 (48 nm) using an electron beam evaporator at temperature of 100 °C. The DBR reflectivity at the wavelength of 451 nm increased with an increase of the number of n-pairs. Although the highest DBR reflectivity at the wavelength of 451 nm was 98.6% as n=6, the DBR reflectivity at the wavelength of 550 nm and 625 nm was 31.9% and 15.8%, respectively, which adversely affected the emitting of the green light. The reflectivity of the (TiO2/SiO2)5-pairs/TiO2 DBR structure at the wavelength of 451 nm, 550 nm, and 625 nm was 96.4%, 13.7%, and 7.6%, respectively. Consequently, the 5-pairs (TiO2/SiO2) were more suitable for the DBR structure than the 6-pairs (TiO2/SiO2).
In this work, we used the green and red QDs as the color conversion layer to transfer the blue light to green and red lights, respectively. For the green and red QDs/micro-LEDs, the bottom HBR structure was designed to reflect the downward emitted lights and the top DBR structure was designed to reflect the non-absorbed blue light that wasn’t absorbed by the QDs, which could increase the probability of the QDs excited by the reflected blue light. Otherwise, the blue light absorption material was deposited on the DBR to absorb the blue light escaped from the DBR, which could improve the color purity of the resulting green and red QDs/micro-LEDs. The electroluminescence spectra of the resulting blue micro-LEDs and the resulting QDs/ micro-LEDs were measured to verify the function of the HBR, DBR and blue light absorption layer. Figure 6 shows the electroluminescence spectra of the resulting green and red QDs/micro-LEDs and the blue micro-LEDs with HBR. As shown in Figure 6, the output intensity of the green and red QDs/micro-LEDs with HBR was respectively increased 11% and 10% in comparison with the QDs/micro-LEDs without HBR. It proved that the HBR could reflect the downward emitted red, green, and blue lights. Furthermore, it was worth noting that only a partial of blue light was absorbed by the green and red QDs, which lead to non-absorbed blue light would blend with the green light and red light and simultaneously emitted from the green and red QDs/micro-LEDs with HBR. Therefore, the non-absorbed blue light emitted from the QDs/micro-LEDs with HBR could be effectively reflected to the QDs color conversion layer by using the DBR with high reflectivity at wavelength of 451 nm and excite more red and green lights. Compared with the QDs/micro-LEDs with HBR and without DBR, the output intensity of the green and red QDs/micro-LEDs with HBR and DBR was respectively increased 20% and 23%. However, since the reflectivity of the DBR was not 100% and the reflectivity of the DBR was decreased as the light was not perpendicularly incident into the DBR structure, a small portion of blue light was stilly escaped, which would affect the color purity of the QDs/micro-LEDs. In this work, the blue light absorption layer was deposited on the DBR to reduce the escaped blue light. The absorptivity spectrum of the blue light absorption layer was shown in Figure 7. The absorptivity of the blue light absorption layer at wavelength of 451 nm, 550 nm, and 625 nm was 77.2%, 1.3%, and 1.8%, respectively. Since the blue light escaped from the DBR could be absorbed by the blue light absorption layer, the intensity of the blue light for the QDs/micro-LEDs was reduced as shown in Figure 6, which could improve the red and green color purity.
Reviewer 4 Report
Improved Color Purity of Monolithic Full Color Micro-LEDs Using Distributed Bragg Reflector and Blue Light Absorption Material
Manuscript No. coatings-743269
This manuscript describes an RGB micro-LED array where red and green color conversion is achieved by the use of CdSe/ZnS core-shell quantum dots. The emission intensity of each color pixel was increased and its color purity (saturation) increased by the use of DBRs and a blue light absorbing material. This work is interesting, useful and written in good English. However, a few questions need to be answered before the manuscript could be considered for publication acceptance.
1.What exactly is the distinct advance being reported here? Micro-LEDs, color conversion in micro-LEDs using QD materials and use of DBRs have all been reported before in many publications. The authors make it sound (although they do quote other publications) as if this is more-or-less a new device. There is only some aspect which is actually new. Is this the use of the blue light absorbing material? Integration of the various components as mentioned in this manuscript? This needs to be very clearly stated.
In this work, the main improvement was higher color purity and joined external structures to make the color conversion efficiency increased. The methods are as follows:
(1) The black photoresist was used as the light-shielding structure to reduce the side leakage.
(2) The functions of the HBR included high reflectivity at the regions of the blue, green, and red lights, and high reflectivity at omnidirectional. Consequently, the HBR deposited on the bottom side of the micro-LEDs to reflect the downward blue, green, and red lights, which could improve the output light intensity.
(3) The distributed Bragg reflector (DBR) deposited on the top of color conversion layer to reflect the non-absorbed blue light that wasn’t absorbed by the QDs, which could increase the probability of the QDs excited by the reflected blue light. Consequently, the color conversion efficiency of the QDs/micro-LEDs could be improved.
(4) The blue light absorption layer was deposited on the DBR to absorb the escaped blue light from the DBR, which could improve the color purity of the QDs/micro-LEDs.
According to the above structure designs, the performance of the color conversion efficiency and color purity for the QDs/micro-LEDs with HBR, DBR, and blue light absorption layer.
2. Some experimental details have not been provided. Such as, what was the black photoresist that was used?
The black photoresist (Model: ABK408X) was supported by Daxin Materials Corp., Taiwan. Currently, the black photoresist can be applied to the color filters of liquid crystal displays as the partition wall. Since it has high light shielding in the visible light, it can prevent crosstalk among the full color micro-LEDs.
3. Why have the green and blue QD spectra been measured with excitation from a tunable laser? Why not use just the blue LED itself as the excitation source?
After we bought the green and red QDs, the photoluminacence (PL) and the absorptance characteristics of the green and red QDs were checked firstly. In our PL measurement system, the excited sources include the He-Cd laser and tunable laser (Optotek Co.). We can not add the blue LED into our PL measurement system as the excited source. Consequently, we used the tunable laser with a wavelength of 451 nm as the excited source.
4. The FWHM of the blue, red and green emission shown in figures 2(a) and 2(b) need to be mentioned. The FWHM of the green and red QD emissions look quite broad for QD emission. Aren’t QDs used in such application to obtain narrow emissions compared to what would be possible with phosphors?
The full width at half maximum (FWHM) of the blue, green and red lights emitted from the blue micro-LEDs, red and green QDs micro-LEDs was 27.0, 32.2, 34.5 nm, respectively. The sentence was added in p. 3, line 107-108 in the revised manuscript.
The green QDs (Model: A540, FWHM<35 nm) and red QDs (Model: A620, FWHM<35 nm) was supported by Taiwan Nanocrystals Inc., Taiwan. In the near future, we try to find and buy the green and red QDs with narrow emission FWHM.
5. In figure 1 I can see the p ohmic contact for the blue pixel but not for the red and green pixels. Where are they? If these are buried under the DBR and QD layers then how are these electrically accessed?
Figure 1(b) shows the top view photographic of the blue micro-LEDs. As shown in the Figure 1(b), since we designed an extended p-electrode in the micro-LEDs, we can measure the electrical characteristic of the resulting QDs/micro-LEDs by external extended p-electrode. We added the top view photographic of the blue micro-LEDs as the Figure 1 (b) in the revised manuscript.
Reviewer 5 Report
In their manuscript "Improved Color Purity of Monolithic Full Color Micro-LEDs ..." Shao-Yu Chu and coworkers describe the design of a full color Micro-LEDSs. They show the influence of various design components on the final performance of the device. The device, the improvements, and the final results are well described and for sure of interest for the readers of the journal "Coatings". The only important points I have to ask are
1. The authors should add an electron microscope image of the device, and not only show a schematic.
At present, since we can’t obtain the SEM or TEM image of the resulting QDs/micro-LEDs, please allow us do not show the SEM or TEM image in the revised manuscript. We added the top view photographic of the blue micro-LEDs as the Figure 1 (b) in the revised manuscript.
2. The authors should add a more detailed description (or give a reference) how they measured the optical properties (type of spectrometer etc. )
In this work, an electroluminescence system was installed in a black box and was equipped with an Agilent 4156C semiconductor parameter analyzer, an optical fiber and spectrometer (Ocean Optics USB2000) system. In the EL measurement process, the resulting samples were placed on the stage in the black box and the optical fiber was fixed above the micro-LEDs with a distance of 1 mm to catch the emitting light from the micro-LEDs. The maximum EL intensity of all the micro-LEDs applied current of 1.2 mA was measured by adjusted the sample stage.
These sentences were added in p. 3, line 98-104 in the revised manuscript.
3. Line 43/44: "... using blue light emitted from the gallium nitride-based micro-LEDs excited the various QDs dimensions." Is this sentence correct? Maybe the authors can reformulate it a bit.
Thanks reviewer’s comments. The sentence was rewritten as "To overcome the above mention problem, the blue light emitted from the gallium nitride (GaN)-based micro-LEDs were used to excited the QDs with various dimensions to form the monolithic red and green quantum dots (QDs)/micro-LEDs [5-7]."in p.1, line 42-44 in the revised manuscript.
4. Line 154: "... a small portion of blue light was stilly escaped." Maybe the authors mean "... a small portion of lbue light still escaped."?
Thanks reviewer’s comments. The sentence was rewritten as “a small portion of blue light still escaped.” in p. 6, line 165-166 in the revised manuscript.
Round 2
Reviewer 2 Report
1.The title needs to be changed to reflect that the main originality of this work is blue light absorbing material not distributed Bragg reflector (DBR). DBR has already been reported in the author's previous work.
Thanks reviewer’s commons. Since the distributed Bragg reflector (DBR) played an important role for reflecting the non-absorbed blue light that wasn’t absorbed by the QDs, please allow us keep the distributed Bragg reflector in the manuscript title.
2. Why the optimized thickness is determined by d = lambda/(4n) ? I believe a theoretical optimization based on Fresnel equation is necessary here.
The DBR mirror is composed of two materials with large differences refractive index. In this work, we selected the TiO2 and the SiO2 as the materials with high refractive index of 2.31 and low refractive index of 1.46 at wavelength of 451 nm, respectively. Figure R1 shows the schematic configuration of the DBR. We referenced Kasap,S.O. Optoelectronics and Photonics: Principles and Practices, 2nd ed.; Pearson Education Inc.: New Jersey, NJ, USA, 2013; pp. 56-58. [Ref 1] to estimate the thickness of the DBR. In order to get the DBR with high reflectivity at some specific wavelength, the reflected lights from each interface of the DBR must be constructive interference. Therefore, the phase difference between the reflected lights must be an integer multiple of 2p. The optical path difference L and the phase difference d of the reflected lights are expressed by a formula as where d and n are the thickness and refractive index of the film, respectively, and k=2p/l is the wave number of the reflected lights. In Figure R1, to form the constructive interference, the phase difference between the reflected lights (RL1, RL2, and RL3) is an integer multiple of 2p. In addition, when the light in the TiO2 layer with high refractive index reflected at the TiO2-SiO2 boundary, there was no phase change in the reflection light. However, the light in the SiO2 layer with low refractive index reflected at the SiO2-TiO2 boundary, there is a p phase change in the reflection light. Consequently, the phase difference d of the reflected lights should be an odd multiple of p. According to the above description, we could rewrite Eq. (2).The thickness of the layers in the DBR structure could be derived from Eq. (3) and shown as follow.
3. The author mentioned that the thickness of TiO2 and SiO2 was further optimized in this work. What kind of thickness optimization have you done in this work ? It seems that the authors only optimized the number of pairs, but still using exactly the same thickness as in their previous work.
We used an electron-beam evaporator system with an optical reflection monitoring system (Asahi Spectra Co., Ltd., model:1200A) to deposit the DBR structure. Firstly, we respectively deposited the TiO2 layer and the SiO2 layer on the sapphire substrate to measure the refractive index of TiO2 and SiO2. After that, we set the reflected wavelength of 451 nm in the optical reflection monitoring system before depositing the DBR structure. In the alternate deposition processes of the TiO2 and SiO2 layers, we could immediately observe the reflectivity signal from the optical reflection monitoring system. We measured the reflectivity of the DBR structure using the UV-Visible spectrophotometer. Finally, we feedbacked the measured reflectivity and revised the deposited thickness of the TiO2 and SiO2 layers to obtain the DBR with higher reflectivity at the wavelength of 451 nm. This method could achieve the optimization of the film thickness.
4. The author mentioned that the measurement results in Figure 4 and 5 match with the theoretical values. Please provide theoretical calculations to support this claim.
To find the optimized condition for the DBR, the electron-beam evaporator system with optical reflection monitoring system was used to deposit the DBR on the sapphire substrate. The theoretical maximum reflectance of the DBR structure with N-pairs could be estimated by the formula as follows: [Ref 1,2]
where nsub of 1.76, nTiO2 of 2.31, and nSiO2 of 1.46 are the refractive index of sapphire substrate, TiO2 layer, and SiO2 layer at wavelength of 451 nm, respectively. Consequently, the theoretical maximum reflectance of the DBR with 5-pairs was 97.7% at the wavelength of 451 nm. In our work, the measured reflectivity was 96.4% at the wavelength of 451 nm. The difference between the theoretical maximum reflectance and the measured reflectivity was 1.3%. Consequently, the measured reflectivity of the DBR was similar with the theoretical maximum reflectance.
[Ref 2] Sheppard, C. J. R. Approximate calculation of the reflection coefficient from a stratified medium. Pure Appl. Opt. 1995, 4, 665–669.
Reviewer 3 Report
I careful read the new version of the manuscript and the author's comment.
Unfortunately, I didn't find a solution to the questions risen in the previous report.
English form: a couple of examples:
errors in the expression - "a problem can be" solved, not "improved" as written in the manuscript;
errors in grammar - "... measurements by adjusted ..." instead, I suppose, "adjusting".
Many other similar cases can be found in the manuscript. I am aware that a fluent english redaction can be difficult for non native speaking people, but the effort to avoid misunderstanding and to clearly expose the concepts should be done in the interest of authors themselves. Otherwise the reading of the paper becomes very difficult.
Physics: It is likely that the choice of a DBR with 5 repeated units is the correct one, but the evidence given in Figure 5 and the deduced reflectance values are questionables. The behavior of the interference fringes for wavelengths between 500 and 800 nm (at the right of the photonic bandgap) exhibits a well assessed trend as a function of layers number, either in terms of minima- and maxima- spectral position, either in terms of reflectance intensity (at maxima and minima). This can be also verified by a simple simulation. In Figure 5 this is no more true precicely for n=5. Why ?
Is the measurement wrong ? or is the model or whatsoever changed the structure with respect to the other ones ?
I admit that the choice of n=5 can be correct, and the position of interference fringes could support that, but the numerical values have some problem and this has not been considered nor justified.
Consistency: the sentence: " As shown in Figure 6, the output intensity of the green and red QDs/micro-LEDs with HBR was respectively increased 11% and 10% in comparison with the QDs/micro-LEDs without HBR. "
is false just because in Figure 6 only data "with HBR" are shown.
It is a very naive question. What is it wrong ? the sentence or e.g. the legend of the figure ?
The reported flaws are indicative of a superficial and not careful preparation of the manuscript. I hope that this is not reflecting the quality of the presented data, but it creates confusion and misunderstanding in the reader.
Then, I can't but confirm the previous report.
Thanks reviewer’s commons. We checked the manuscript and revised some errors in the revised manuscript.
The DBR mirror is composed of two materials with large differences refractive index. In this work, we selected the TiO2 and the SiO2 as the materials with high refractive index of 2.31 and low refractive index of 1.46 at wavelength of 451 nm, respectively. Figure R1 shows the schematic configuration of the DBR. We referenced Kasap,S.O. Optoelectronics and Photonics: Principles and Practices, 2nd ed.; Pearson Education Inc.: New Jersey, NJ, USA, 2013; pp. 56-58. [Ref 1] to estimate the thickness of the DBR. In order to get the DBR with high reflectivity at some specific wavelength, the reflected lights from each interface of the DBR must be constructive interference. Therefore, the phase difference between the reflected lights must be an integer multiple of 2p. The optical path difference L and the phase difference d of the reflected lights are expressed by a formula.
where d and n are the thickness and refractive index of the film, respectively, and k=2p/l is the wave number of the reflected lights. In Figure R1, to form the constructive interference, the phase difference between the reflected lights (RL1, RL2, and RL3) is an integer multiple of 2p. In addition, when the light in the TiO2 layer with high refractive index reflected at the TiO2-SiO2 boundary, there was no phase change in the reflection light. However, the light in the SiO2 layer with low refractive index reflected at the SiO2-TiO2 boundary, there is a p phase change in the reflection light. Consequently, the phase difference d of the reflected lights should be an odd multiple of p. According to the above description, we could rewrite Eq. (2).
The thickness of the layers in the DBR structure could be derived from Eq. (3) and shown as follow.
To find the optimized condition for the DBR, the electron-beam evaporator system with optical reflection monitoring system was used to deposit the DBR on the sapphire substrate. The theoretical maximum reflectance of the DBR structure with N-pairs could be estimated by the formula as follows: [Ref 1,2], where nsub of 1.76, nTiO2 of 2.31, and nSiO2 of 1.46 are the refractive index of sapphire substrate, TiO2 layer, and SiO2 layer at wavelength of 451 nm, respectively. Consequently, the theoretical maximum reflectance of the DBR with 5-pairs was 97.7% at the wavelength of 451 nm. In our work, the measured reflectivity was 96.4% at the wavelength of 451 nm. The difference between the theoretical maximum reflectance and the measured reflectivity was 1.3%. Consequently, the measured reflectivity of the DBR was similar with the theoretical maximum reflectance.
Reviewer 4 Report
The authors have provided satisfactory answers to all questions and have made required corrections/additions to the manuscript. The revised manuscript is now in a suitable form for publication in Coatings. I would be happy to see this interesting article in print.
Thanks reviewer’s comments and affirmations. Your suggestion makes the article more complete.
Round 3
Reviewer 2 Report
All my comments have been addressed.